engineering geology

water permeability, shale, triaxial testing, fracture, permeability model

**Author for correspondence:**
Dongming Zhang
e-mail: zhangdm@cqu.edu.cn

# Triaxial testing on water permeability evolution of fractured shale

## Menglai Wang[1] and Dongming Zhang[2]

[1]National Engineering and Technology Center for Development and Utilization of Phosphate Resources, Yunnan phosphate group Co., Ltd, Kunming, Yunnan, People's Republic of China
[2]State Key Laboratory of Coal Mine Disaster Dynamics and Control, School of Resources and Safety Engineering, Chongqing University, Chongqing, People's Republic of China

DZ, 0000-0002-6303-8829

A sound understanding of the water permeability evolution in fractured shale is essential to the optimal hydraulic fracturing (reservoir stimulation) strategies. We have measured the water permeability of six fractured shale samples from Qiongzhusi Formation in southwest China at various pressure and stress conditions. Results showed that the average uniaxial compressive strength (UCS) and average tensile strength of the Qiongzhusi shale samples were 106.3 and 10.131 MPa, respectively. The nanometre-sized (tiny) pore structure is the dominant characteristic of the Qiongzhusi shale. Following this, we proposed a pre-stressing strategy for creating fractures in shale for permeability measurement and its validity was evaluated by CT scanning. Shale water permeability increased with pressure differential. While shale water permeability declined with increasing effective stress, such effect dropped significantly as the effective stress continues to increase. Interestingly, shale permeability increased with pressure when the pressure is relatively low (less than 4 MPa), which is inconsistent with the classic Darcy's theory. This is caused by the Bingham flow that often occurs in tiny pores. Most importantly, the proposed permeability model would fully capture the experimental data with reasonable accuracy in a wide range of stresses.

## 1. Introduction

The exploitation of unconventional gas is expected to slow down the consumption of coal resources. China has rich shale gas resources, the US Energy Information Administration (EIA) noted that the technically recoverable shale gas reserves in China are 1115 trillion cubic feet (TCF), ranking the first across the globe [1]. Unconventional gas mainly includes shale gas, tight gas and coalbed methane. To date, shale gas is the most promising hydrocarbon resource among the three since

commercial shale gas production at economic cost has been achieved in the United States, Canada and China [2,3]. The huge success in shale gas development has triggered other countries to explore and produce shale gas. However, the pore network of shale is complex with sizes ranging from a few nanometres up to several micrometres, making the rock less permeable (less than $10^{-18}$ m$^2$), which is the main challenge for shale gas production [4]. In order to enhance the fluid transport capacity and improve the permeability of shale reservoirs, hydraulic fracturing technology is often applied. Hydraulic fracturing technology was first used in the petroleum industry, its main purpose is to develop complex fracture networks for providing channels for gas transport, and therefore increasing the gas flow rate [5–7]. However, due to the limitation of experimental set-ups, it is difficult to test the water permeability during the fracturing process in the laboratory (for example, how to directly test the post-fracturing water or gas permeability individually) and the behaviour of water transport in shale fractures still remains unclear. Instead, investigating the permeability evolution of fractured shale would be an effective way to resolve this concern and has gained wide attention in the unconventional resource industry and academia [8].

To date, limited studies have reported the water permeability of shale at various conditions. Van Noort & Yarushina [9] conducted 43 permeability measurements using water as the fluid on a shale sample from Svalbard. It is found that permeability decreases with increasing confining pressure due to instantaneous, time-dependent and permanent compaction. Qu et al. [10] studied the fluid flow in shale tensile fracture, simulated the fluid flow in shale fracture with opening of 0.02–0.40 mm by lattice Boltzmann method, and developed an equation to calculate the permeability of shale with tensile fracture. Dong et al. [11] tested the permeability of shale from Taiwan Chelungpu fault drilling project (TCDP) Hole-A, reporting that reservoir interaction and stress change have great influence on rocks, especially low permeability rocks like shale. Carey et al. [12] investigated the permeability characteristics of fractured shale and found that the activation of existing fractures is more likely to be the source of permeability for rocks with fractures under compression. It can be clearly seen that few studies have reported the water permeability evolution of fractured shale.

In the present study, we firstly developed a pre-stressing strategy to obtain fractured shale samples, according to the uniaxial compressive strength (UCS) and tensile strength based upon uniaxial compression test and Brazilian tension test. Following this, we drew upon the GCTS test system to measure the water permeability of six shale samples at various pressure and stress conditions. Also, the development of randomly propagated fractures before and after the permeability measurement was evaluated by CT scanning (three-dimensional reconstruction); and the pore size distribution (PSD) of shale was detected using nuclear magnetic resonance (NMR) technique. Finally, and most importantly, we discussed the pressure and stress dependence of water permeability and applied a proposed model to match the permeability data in our study. The present study aims to provide theoretical guidance for evaluating and predicting the water permeability of shale after fracturing, especially the water permeability evolution with stress measurements in steps.

# 2. Geological setting

The Yangtze platform is one of the three oldest platforms in South China, which is divided into upper, middle and lower parts. The basement of the Yangtze platform is mainly composed of Neoproterozoic low-grade metamorphic rocks that were formed in the Palaeozoic geological period. After the Early Palaeozoic, marine sediments began to cover the platform, including extremely thick sedimentary shale formations [13]. Along the southern margin of the Yangtze platform, a set of Early Cambrian black shale sequences are deposited, mainly consisting of shale, mudstone and siliceous rocks. At present, it is generally believed that the formation environment of the lower Cambrian black shale experienced a process from the hydrostatic reducing environment to the secondary oxidation reducing environment [14–16].

According to the systematic regional geological survey in the last century, there are thick black rock series in the lower Cambrian, lower Silurian and upper Silurian in eastern Yunnan province, China, which are the main source rocks for the shale gas system. There is a consensus that the shale reservoir in Qiongzhusi Formation is mainly composed of clay minerals with an average content of 50.2%, most of which is illite; while the average content of brittle minerals is 29.85%, most of which is quartz. Also, previous literature has reported the total organic carbon of Qiongzhusi shale is 1.42–2.51% (organic-rich), and the equivalent vitrinite reflectance ($R_o$) or maturity level is 2.16–3.32% [17]. Table 1 lists the tested mineralogical composition of the shale sample using X-ray diffraction analysis. Both

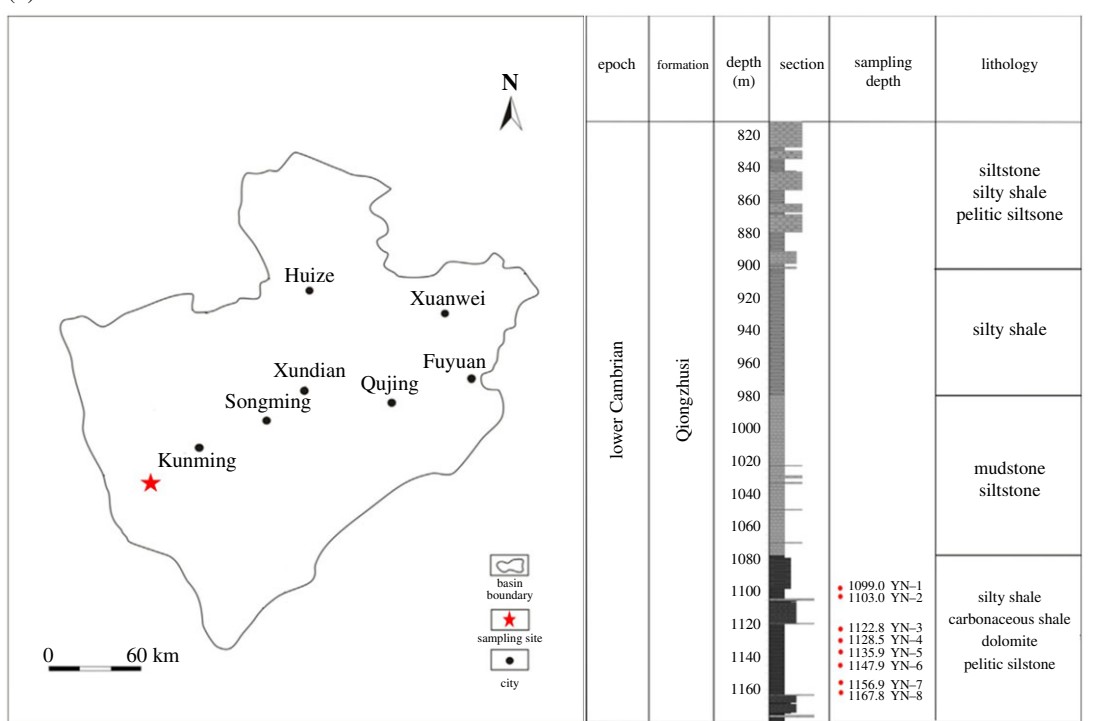

**Figure 1.** (*a*) Geographical location and (*b*) stratigraphical units of the Qiongzhusi Formation [17].

**Table 1.** Mineralogical composition (%) of the Qiongzhusi shale sample.

| | | | | | | clay | |
|---|---|---|---|---|---|---|---|
| sample | quartz | dolomite | feldspar | pyrite | calcite | illite | chlorite |
| | 38.3 | 7.5 | 5.5 | 2.6 | 7.8 | 29.9 | 8.4 |

the previous data and table 1 demonstrate the huge potential for promising gas reserves in southwest China, especially the Qiongzhusi Formation. Additionally, the geographical location and stratigraphic units of the Qiongzhusi Formation are plotted in figure 1.

# 3. Sample characterization and methods

## 3.1. Sample characterization

We drilled the shale blocks from the west bank of Dianchi Lake in the southwest of Kunming City (figure 1), which belongs to the lower Cambrian sedimentary rocks in eastern Yunnan. Then the shale blocks were cut and ground using a coring machine in the laboratory to obtain several shale core samples with a height of 100 mm and a diameter of 50 mm. The prepared six shale samples (nos. 1–6) for permeability measurement are shown in figure 2.

## 3.2. Experimental methods

The whole permeability measurement was conducted on the GCTS testing system at the State Key Laboratory of Coal Mine Disaster Dynamics and Control, Chongqing University. It is able to perform experimental testing on mechanical properties and permeability characteristics of rock under *in situ* stress conditions. As for the technical parameters of the GCTS testing system, the maximum axial load is 3000 kN, the maximum confining pressure is 200 MPa, the maximum pore water pressure is

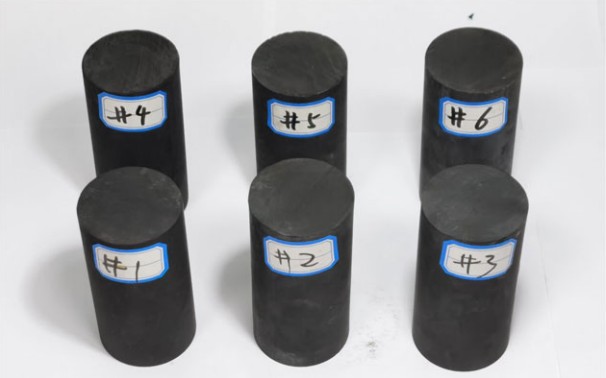

**Figure 2.** Prepared six shale samples (nos. 1–6) for permeability measurement.

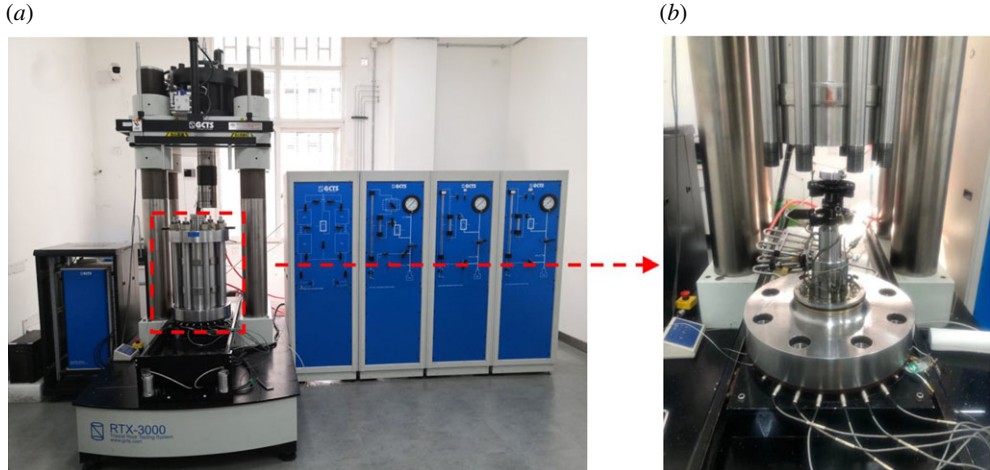

**Figure 3.** (*a*) GCTS testing system and (*b*) its triaxial cell.

200 MPa, and the maximum temperature is 200°C. Figure 3 shows a comprehensive illustration of the GCTS testing system.

Given the advantages of time saving and a straightforward solution, we used the steady-state method to measure the water permeability of shale samples, and therefore the permeability could be calculated according to Darcy's law [18]

$$k = \frac{\mu L Q}{A(P_1 - P_2)},$$

(3.1)

where $k$ is the permeability ($m^2$), $\mu$ is the viscosity of the permeating fluid (MPa s), $L$ is the length of the sample (m), $Q$ is the flow rate ($m^3\ s^{-1}$), $A$ is the cross-sectional area of the sample ($m^2$), $P_1$ is the inlet pressure (MPa) and $P_2$ is the outlet pressure (MPa).

To ensure that the permeability is measured in fractured shale, the extremely tight shale samples should be processed, prior to the permeability measurement, to artificially create fractures inside the samples. Previous studies often used a single pseudo-fracture by splitting the shale sample to measure the permeability, which leads to several questionable issues [19,20]. For example, the shale sample after splitting is no longer an intact rock, and the real micro-fractures after hydraulic fracturing should be randomly distributed with numerous secondary fractures. Hence, pre-stressing the samples to create more reliable fractures in shale for permeability measurement is needed. In this regard, a better understanding of the shale strength properties, such as UCS and tensile strength, is essential to meet the requirements. The maximum axial stress for pre-stressing the sample should be less than UCS (often two-thirds of UCS), to effectively create randomly distributed micro-fracture, but not to break the sample. Whereas the injecting water pressure during the permeability measurement should be less than the tensile strength of the sample, to ensure that no extended or developed fractures (fracture

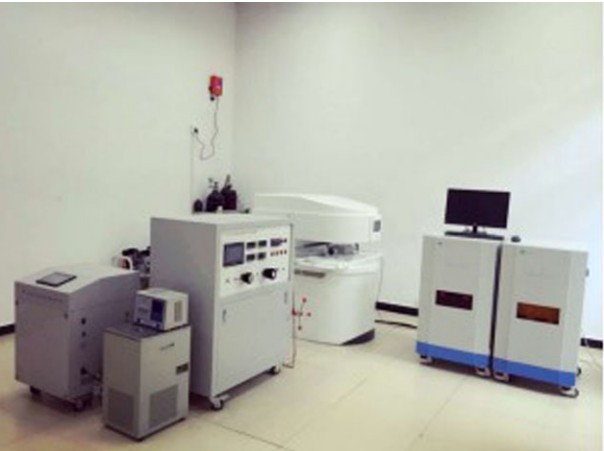

**Figure 4.** MacroMR12-150H-I NMR analyser.

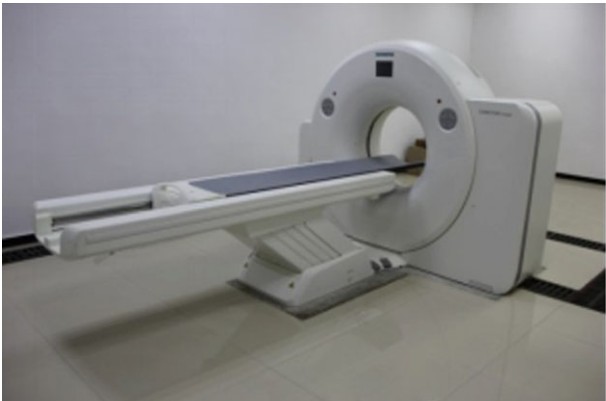

**Figure 5.** SOMATOM Ccope CT scanning system (SIEMENS).

extends, and a relatively developed pore network and randomly distributed cracks have been formed) are newly formed, making the measured permeability to be comparable. In this study, we used the MTS815 rock mechanics test system and rock compression shear test machine (Brazilian tension test) to measure the UCS and tensile strength of shale samples, so as to understand the strength characteristics of shale samples for designing an optimal strategy to create reliable fractures in shale when pre-stressing the samples before the permeability measurement.

The unique ability of NMR makes it a key and novel laboratory analytical tool for shale research since NMR technique has the advantage of being both non-intrusive and non-destructive [21–23]. In the present study, we used the MacroMR12-150H-I analyser (figure 4) to detect the internal structure and PSD of the shale sample. Sample no. 1 (100 mm in height and 50 mm in diameter) was selected for NMR measurement.

Last but not least, we used the SOMATOM Ccope CT scanning system (figure 5) to see the fracture development before and after the permeability measurement. The image processing and three-dimensional CT reconstruction treatment were performed using the Avizo software. Also, sample no. 1 (100 mm in height and 50 mm in diameter) was selected for CT scanning.

## 3.3. Experimental design for permeability measurement

To investigate the water permeability evolution of fractured shale, we designed several stress conditions to measure the water permeability of shale. The entire testing process was operated at the constant temperature as 25°C, and the effect of temperature on permeability evolution of shale is neglected in this study (the fixed temperature is to ensure that isothermal conditions were applied to avoid undesirable variations in flow rate and pressure). Also, the outlet pressure was kept constant as atmospheric pressure (0.1 MPa), and fluid flow was along the axial direction. Table 2 summarizes the designed stress states to measure the permeability of shale samples.

**Table 2.** Summary of the designed stress states to measure the permeability of shale samples.

| confining pressure (MPa) | axial stress (MPa) | inlet pressure (MPa) |
| --- | --- | --- |
| 1 | 10 | 1.0 |
| | | 1.2 |
| | | 1.4 |
| | | 1.6 |
| | | 1.8 |
| | | 2.0 |
| 2 | 12 | 1.4 |
| | | 1.6 |
| | | 1.8 |
| | | 2.0 |
| | | 2.2 |
| | | 2.4 |
| 3 | 14 | 1.8 |
| | | 2.0 |
| | | 2.2 |
| | | 2.4 |
| | | 2.6 |
| | | 2.8 |
| 4 | 16 | 2.2 |
| | | 2.4 |
| | | 2.6 |
| | | 2.8 |
| | | 3.0 |
| | | 3.2 |
| 5 | 18 | 2.6 |
| | | 2.8 |
| | | 3.0 |
| | | 3.2 |
| | | 3.4 |
| | | 3.6 |
| 6 | 20 | 3 |
| | | 3.2 |
| | | 3.4 |
| | | 3.6 |
| | | 3.8 |
| | | 4.0 |

# 4. Results

## 4.1. Uniaxial compressive strength results

Additional two shale core samples drilled from the same rock block were tested for determining the UCS of shale sample and relations between the axial stress and strain are plotted in figure 6. It can be seen from the figure that the average UCS of the shale samples is 106.3 MPa. Following this, the maximum

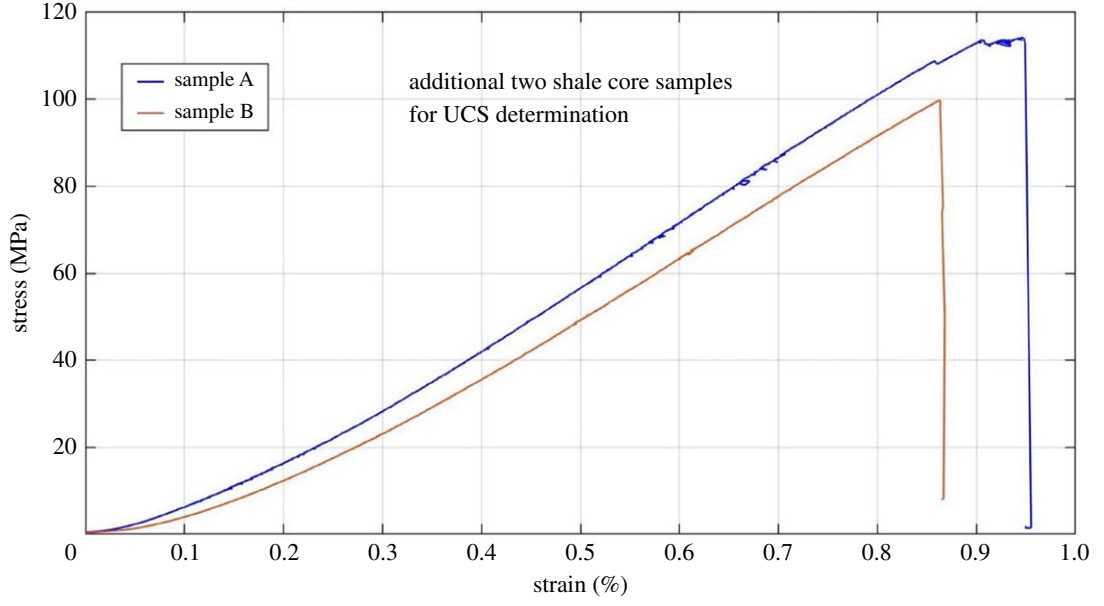

**Figure 6.** Axial stress versus strain of uniaxial compressive test.

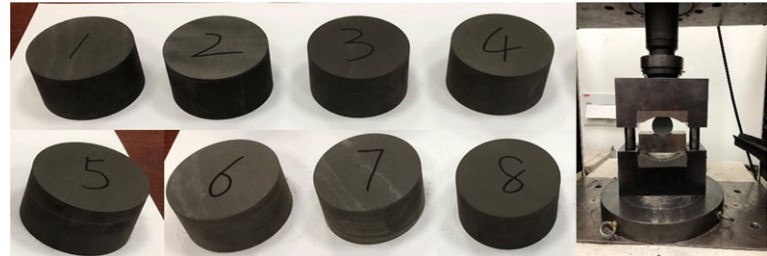

**Figure 7.** Eight shale samples prepared for Brazilian tension test.

axial stress for pre-stressing the six prepared shale samples was set to be 71 MPa (two-thirds of UCS) in our study, to effectively create randomly distributed micro-fracture for permeability measurement.

## 4.2. Brazilian tension test results

A total of eight samples were used in the Brazilian tension test to determine the tensile strength of shale. The length of the sample is 25 mm and the diameter is 50 mm, as shown in figure 7. According to the Brazilian tension test theory, the tensile strength of the sample can be calculated as [24]

$$\sigma_t = \frac{2P}{\pi DL}, \tag{4.1}$$

where $\sigma_t$ is the tensile strength (N), $L$ is the length (thickness) (m), $D$ is the diameter (m) and $P$ is the maximum load (N) when splitting occurs. The splitting curves of the Brazilian tension test are shown in figure 8 and the determined tensile strengths of the samples are listed in table 3. One can report that the average tensile strength is 10.131 MPa, and this is also the reason why we set the inlet pressure values in table 2 (less than 10 MPa) for permeability measurement. Under such conditions, as we mentioned before, no extended or developed fractures can be newly formed, making the measured permeability to be comparable.

## 4.3. NMR results

The most effective way to transform the original NMR $T_2$ spectrum to PSD is to use the following equation [21–23,25]:

$$d = 2\rho_2 F_s T_2, \tag{4.2}$$

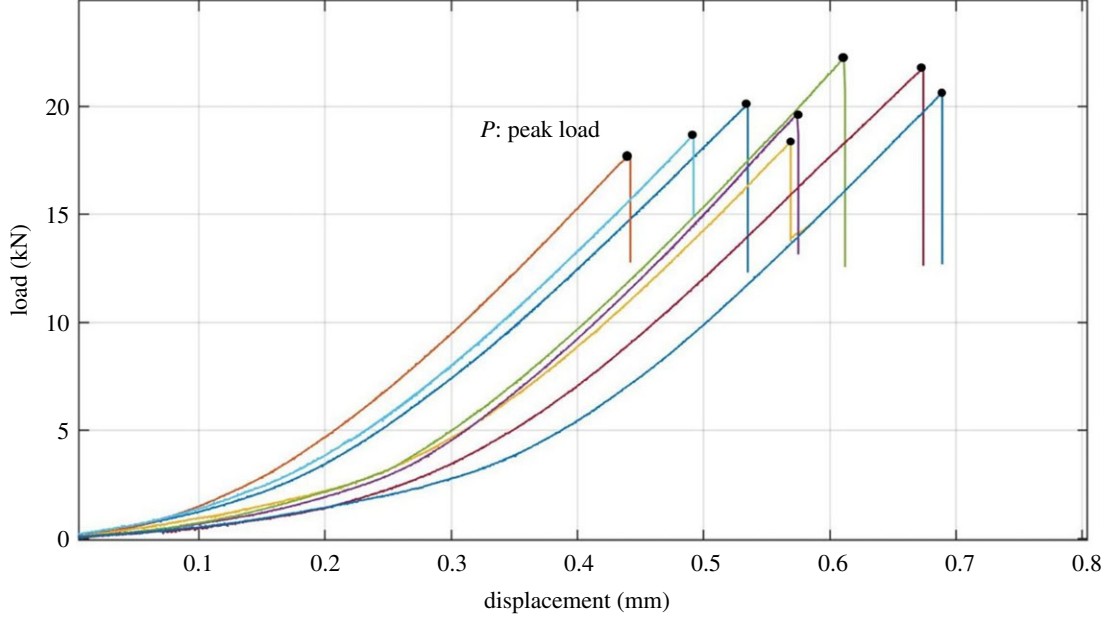

**Figure 8.** The splitting curves of the Brazilian tension test.

**Table 3.** Determined tensile strengths of the shale samples.

| sample | 1 | 2 | 3 | 4 | 5 | 6 | 7 | 8 |
|---|---|---|---|---|---|---|---|---|
| peak load (kN) | 20.078 | 17.617 | 18.351 | 19.671 | 22.296 | 18.679 | 21.765 | 20.671 |
| tensile strength (MPa) | 10.225 | 8.9724 | 9.3464 | 10.018 | 11.355 | 9.5135 | 11.085 | 10.528 |

where $d$ is the pore diameter, $\rho_2$ is the surface relaxivity, $F_s$ is the dimensionless pore shape factor ($F_s = 2$ and 3 for cylindrical pores and spherical pores, respectively; in this work, $F_s = 2$) and $T_2$ is the transverse relaxation time (in this work, $T_2 = 16.25$ nm ms$^{-1}$ [22]). Equation (4.2) shows that short relaxation time represents small pore or medium pore and long relaxation time represents macropore.

The PSD of shale sample no. 1 using NMR technique is plotted in figure 9. It can be seen that the PSD curve of the shale shows obvious three peak characteristics, and shale contains plenty of (95%) nanometre-sized pores (less than 1000 nm) and a small fraction of (5%) macropores. The nanopore structure is the dominant characteristic of shale's unique nature. Additionally, the porosity of the sample is 2.4231%. The influence of shale's internal structure on water transport in shale will be discussed in §5.1.

## 4.4. CT scanning results

The three-dimensional reconstructions of shale's internal structure (sample no. 1) before and after the permeability measurement are shown in figures 10 and 11. As evident from figure 10 showing the slice images at different layers and reconstruction model of shale in the original state, there are no obvious fractures in shale in the original state; while figure 11 shows the slice images and three-dimensional reconstruction model of shale after fracturing treatment. It can be clearly seen from the figure that many obvious fractures appear at the same slice position compared with figure 10. After three-dimensional reconstruction, we find that three penetrating fractures have been formed in the sample. These three main developed fractures in shale as the dominant pathways for fluid (water) transport show that our pre-stressing strategy to create more reliable fractures in shale for permeability measurement has met the requirement.

## 4.5. Permeability results

To investigate how the water permeability of shale evolves with pressure, we plotted the change behaviour of measured permeability with pressure differential (also known as pressure drop) at

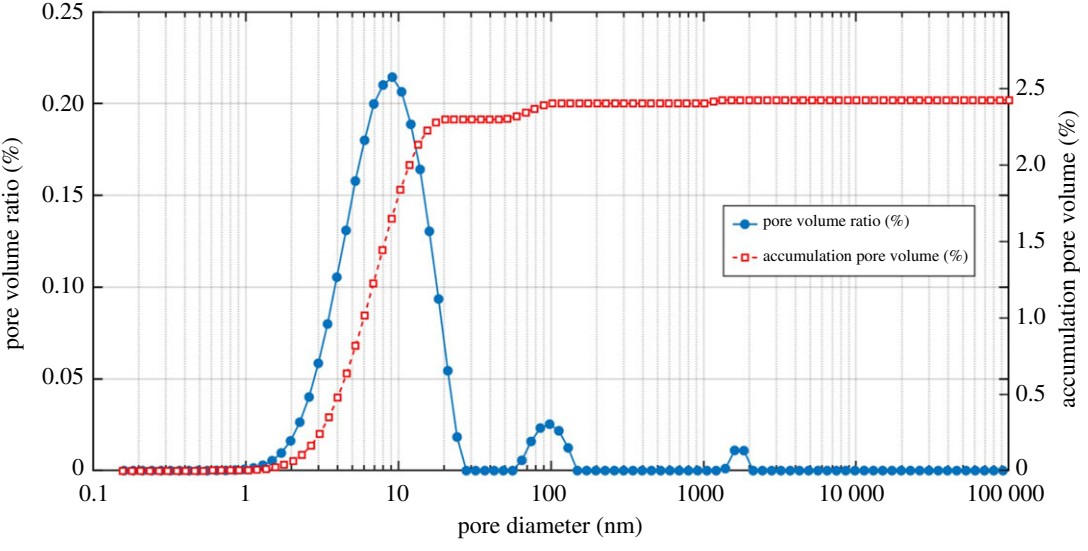

**Figure 9.** Pore size distribution of shale sample no. 1 using NMR technique.

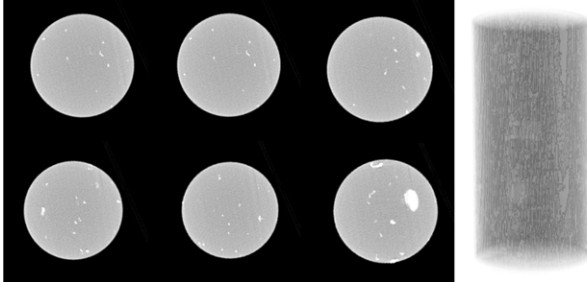

**Figure 10.** Slice images at different layers and three-dimensional reconstruction model of shale (sample no. 1) before the permeability measurement.

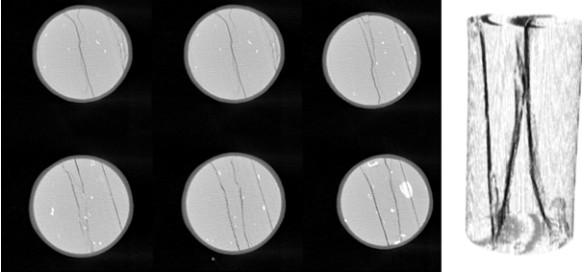

**Figure 11.** Slice images at different layers and three-dimensional reconstruction model of shale (sample no. 1) after the permeability measurement.

different *in situ* stress conditions, according to our experimental design, in figure 12. It can be seen that, no matter how the stress condition varies, the water permeability increased with pressure differential for all the six tested shale samples. In consideration of the fact that the outlet pressure was fixed in our study, it can be inferred that the shale permeability (water) exhibits a positive correlation with pressure or pore pressure. Additionally, figure 12 shows the reduction of shale permeability with increasing deviatoric stress (axial stress minus confining pressure) during the stress loading process. However, permeability demonstrated a vast difference of range and value among the given six deviatoric stresses (stress conditions). In detail, deviatoric stress equaling 9 MPa (figure 12*a*) results in the largest permeability values and widest permeability range (from $1.99 \times 10^{-15}$ to $3.05 \times 10^{-15}$ m$^2$) among the six cases; while a higher deviatoric stress of 14 MPa (figure 12*f*) witnessed the smallest permeability values and narrowest permeability range (from $2.86 \times 10^{-16}$ to $2.94 \times 10^{-16}$ m$^2$). The possible explanation for this

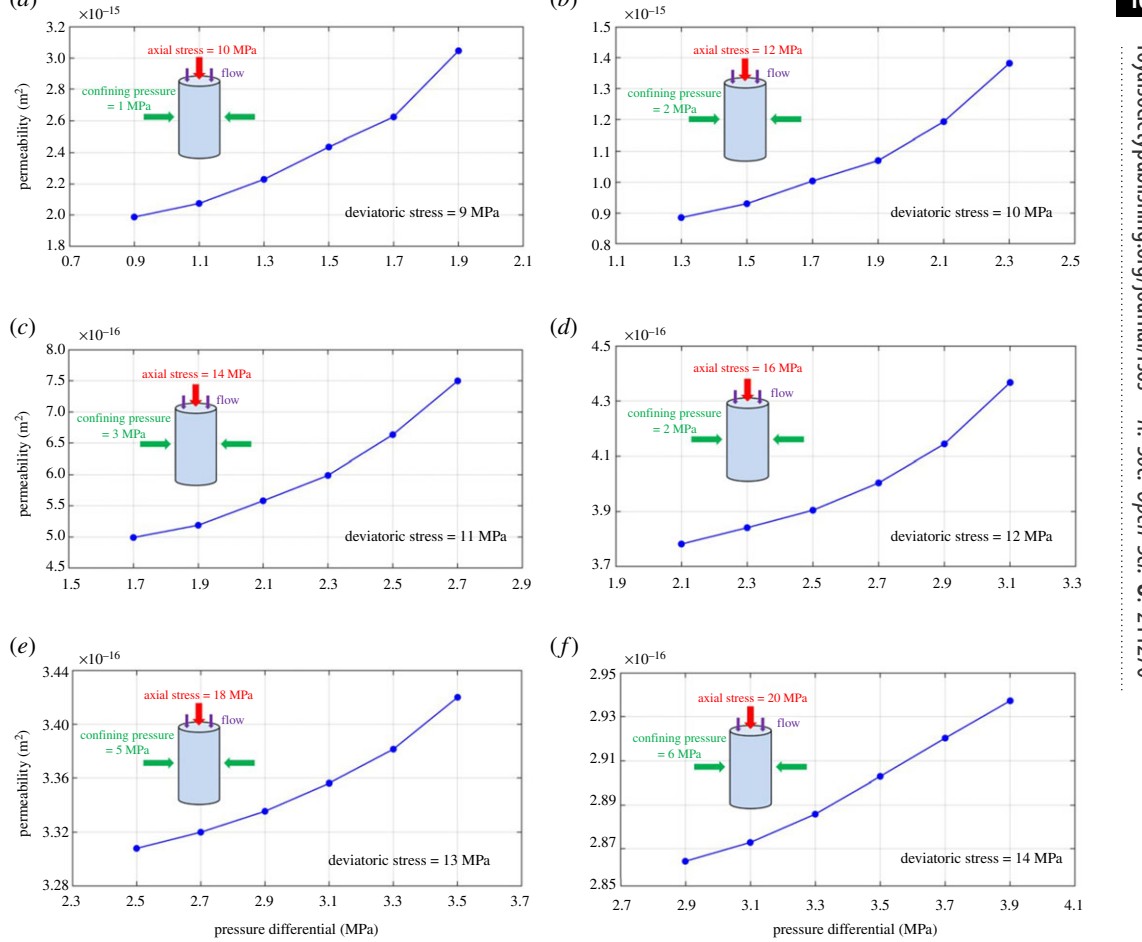

**Figure 12.** Water permeability versus pressure differential at six separate deviatoric stresses.

is that increasing deviatoric stress in this study, in line with our experimental design, leads the effective stress to rise, which compresses the pathways for fluid transport in shale, thus making the permeability to reduce. The pressure and stress dependence of permeability will be discussed in detail hereinafter.

# 5. Discussion

## 5.1. Pressure dependence of permeability

Permeability is an intrinsic property of porous media. That is, it only depends on the internal structure of porous media and is supposed to be independent of fluid properties [26,27]. However, as a tight reservoir rock with unique nature of pore/fracture structure, shale permeability often varies significantly with pressure, which is known as the so-called non-Darcy phenomenon. Previous work reported numerous evidence for gas non-Darcy flow in shale by experimentally testing the gas shale permeability at various pore pressures and stress conditions [18,28–37]. The most dominant contribution to gas non-Darcy flow is the Klinkenberg effect (also known as the gas slippage effect or simply the slip flow), in which the tested apparent permeability is much higher than the intrinsic permeability, causing the non-Darcy phenomenon. The underlying explanation for this interesting finding is that gas molecules collide extensively with pore walls instead of other gas molecules, due to the nanometre-sized pore structure of shale (extremely tight), making gas molecules 'slip' on the pore walls with non-zero velocity (different with liquid flow) and, therefore, the higher permeability (by several times) [38]. Although the gas non-Darcy flow in shale has been reported to some extent in the literature, the amount of previous studies involving the topic as water transport (pressure dependence of water permeability) in shale is still low.

Based upon our experimental data, the pressure dependence of shale water permeability (permeability with pore pressure differential) could be seen in figure 12. It can be concluded that the shale water

permeability varies with pore pressure at all the stress conditions. Specifically, shale permeability increased with pressure when the pressure is relatively low (less than 4 MPa in the present study), which is inconsistent with the classic Darcy's theory, where permeability should remain unchanged with pressure when using Newtonian liquid, e.g. water, as the testing fluid. This inconsistency, to the best of our knowledge, is caused by the Bingham flow (or Bingham effect) that often occurs in tiny pores. According to our NMR measurement data (figure 9), the shale sample contains plenty of (95%) nanometre-sized pores and a small fraction of (5%) macropores; and under such conditions, the Bingham flow would be obvious in shale's tiny pores or micro-fractures. The underlying mechanism for this water non-Darcy flow (or non-Newtonian fluid characteristic) in shale at low pressures (low flow velocity) could be attributed to the existing static friction between the fluid and pore/fracture walls, therefore leading the permeability to reduce. Another hypothetical explanation is that, in consideration of the tiny void space (nanometre-sized pores and some micro-fractures), water could be adsorbed on the inner surfaces of pore walls, which prevents the fluid transport, resulting in a lower permeability. A more in-depth knowledge of the pressure dependence of water permeability (non-Darcy flow) in shale goes beyond the scope of the present work and requires further investigations.

## 5.2. Stress dependence of permeability and permeability modelling

Fluid transport capacity in reservoir rocks is closely related to the stress condition (or applied stress). Hence, a sound understanding of the shale permeability evolution with respect to stress is beneficial to the optimization and strategy-making of commercial shale gas production. The most effective method for investigating the stress dependence of permeability, to date, is to correlate the permeability with effective stress; since it generally shows the compression degree of rock, influencing the pathways (micro-fractures or pores) for fluid flow that controls the permeability evolution [39,40]. In line with the Terzaghi's effective stress theory, we define the effective stress (also called the mean effective stress in other publications) in this study, considering the triaxial stress states, as

$$\sigma_e = \frac{\sigma_1 + 2\sigma_3}{3} - \alpha P_p, \tag{5.1}$$

where $\sigma_e$ is the effective stress, $\sigma_1$ is the axial stress, $\sigma_3$ is the confining pressure, $\alpha$ is the Biot's coefficient (in the present study, $\alpha = 1$), and $P_p$ is the pore pressure.

One of the most significant potential applications of the permeability evolution data is to predict the permeability values at various stresses, and a suitable permeability model is therefore needed. Generally, there is a wide consensus that the permeability evolution with effective stress of reservoir rocks follows a form of exponential (or power) correlation. However, traditional exponential (or power) correlations are often established experimentally only in the presence of confining pressure. Permeability models based on these traditional correlations may not fully capture detailed information when the rock is under *in situ* stress conditions, e.g. triaxial stress, and, therefore, this concern could be resolved by introducing developed models with more mechanisms considered, e.g. fracture compressibility and geometry. In this study, in consideration of the fluid transport in fractured shale, we used a developed exponential-formed permeability correlation derived by [41,42] to model the experimental data. The reason why we used this model is that it is derived based on the fracture compressibility and geometry, consistent with our experimental design and research focus, i.e. fluid transport in fractures. The permeability model takes on the following form:

$$k = k_0 \, e^{-3(c_{f0}/\alpha)(1-e^{-\alpha(\sigma_e - \sigma_{e0})})}, \tag{5.2}$$

where $k_0$ is the initial permeability, $c_{f0}$ is the initial compressibility, $\alpha$ is the declining rate of compressibility and $\sigma_{e0}$ is the initial effective stress. It should be noted that this model is used for gas transport in the previous work and, to the best of our knowledge, the present study is the very first work to apply the model to water (liquid) permeability data of shale.

Changes in water permeability of shale with effective stress and their associated modelling results at six separate deviatoric stresses (according to our experimental design) are plotted in figure 13, and table 4 lists the obtained model parameters. As illustrated in figure 13, shale permeability declined with effective stress for all the experimental cases (by 3 to 50%). This is intuitively understood since increasing effective stress leads the fractures inside the shale sample to be compressed, narrowing the channels for water transport (less permeable). However, such effect dropped significantly with increasing deviatoric stress (as the effective stress continues to increase), for the fact that the range and value of permeability decreased

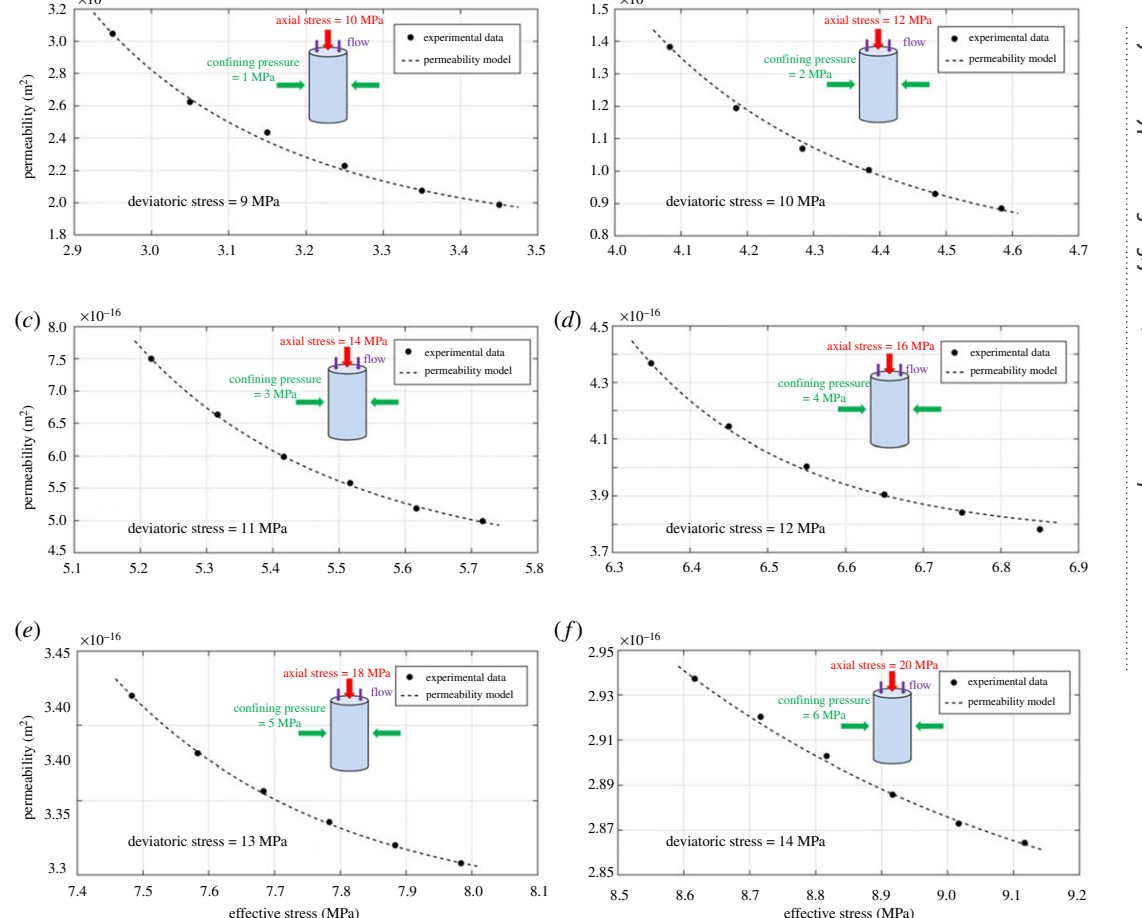

**Figure 13.** Water permeability evolution with effective stress and modelling results at six separate deviatoric stresses. The black solid symbols denote the experimental data and the dotted lines represent the model matching results.

**Table 4.** Model parameters.

| source | deviatoric stress (MPa) | $c_{f0}$ (MPa$^{-1}$) | $\alpha$ (MPa$^{-1}$) | $R^2$ (%) |
|---|---|---|---|---|
| figure 13a | 9 | 0.5458 | 2.9824 | 99.34 |
| figure 13b | 10 | 0.4875 | 2.1229 | 99.56 |
| figure 13c | 11 | 0.4719 | 2.4317 | 99.94 |
| figure 13d | 12 | 0.2284 | 4.1529 | 99.47 |
| figure 13e | 13 | 0.0427 | 2.9968 | 99.94 |
| figure 13f | 14 | 0.0247 | 1.6257 | 99.58 |

dramatically from figure 13a to f. As for the matching performance of the permeability model, one can clearly conclude that the proposed model (equation (5.2)) could fit well with the experimental data with high accuracy at various stress conditions. Interestingly, the matching curve seems to become more linear from figure 13a to f as the effective stress continues to rise. However, it also warrants testing other type of shales to specifically verify this phenomenon. Moving on to the model parameters, figure 14 shows the relation between initial compressibility and deviatoric stress obtained through data matching. It can be concluded that the initial compressibility $c_{f0}$ is positively correlated with deviatoric stress; that is, the initial compressibility decreases with increasing effective stress. This relation is supposed to be reasonable from the scientific point of view, for the fact that shale, as a porous medium, is relatively hard to be compressed when the effective stress is high. Thus, figure 14 also serves more evidence to show the reasonable fit of the proposed permeability model.

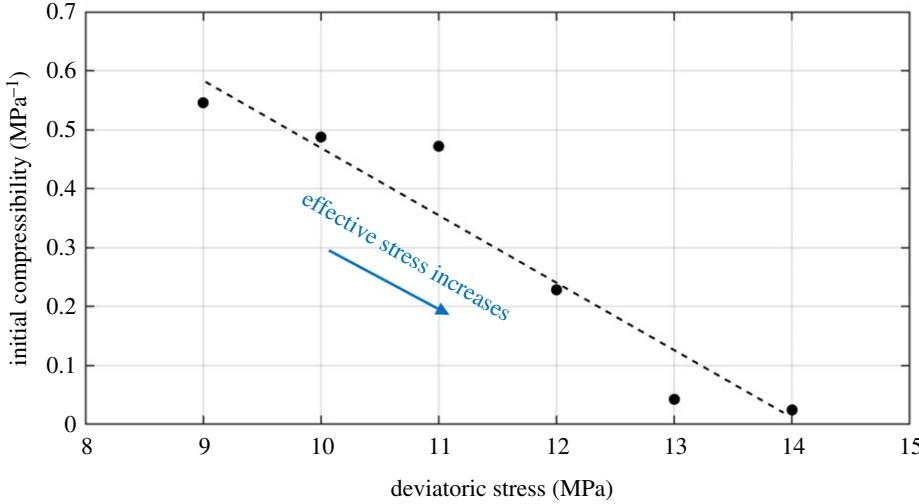

**Figure 14.** Relation between initial compressibility and deviatoric stress obtained through data matching.

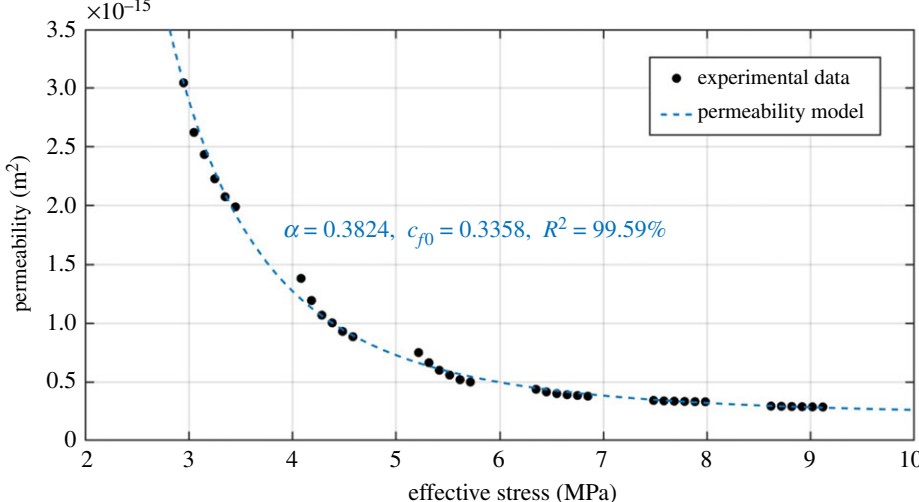

**Figure 15.** Water permeability evolution with effective stress of shale. The black solid symbols denote all the experimental data in this study and the dotted lines represent the model matching results.

To further evaluate the adequacy of the permeability model, we plotted all the permeability data with effective stress in this study and its associated modelling result in figure 15. Again, shale permeability decreased with increasing effective stress for all the experimental data (from $3.05 \times 10^{-15}$ to $2.86 \times 10^{-16}\,m^2$). Most importantly, the proposed permeability model also fully captures the experimental data with reasonable accuracy when using an expanded dataset, demonstrating its validity for a wide range of data. Consequently, we are able to show that the proposed model would predict water permeability of shale at various stress conditions and is beneficial for optimizing reservoir stimulation strategies during the long-time commercial shale gas production.

# 6. Conclusion

We have systematically measured the water permeability of six fractured shale samples from Qiongzhusi Formation at various pressures and stress conditions and discussed some effects on water transport in shale. Based upon our work, the following conclusions can be drawn:

(1) The average UCS and average tensile strength of the Qiongzhusi shale samples are 106.3 and 10.131 MPa, respectively. The nanometre-sized (tiny) pore structure mainly hosted in the matrix is the dominant characteristic of the Qiongzhusi shale.

(2) The fractured shale sample used for permeability measurement was evaluated by CT scanning, showing that the pre-stressing strategy could effectively create randomly distributed fractures in shale.

(3) Shale water permeability increased with pressure differential. While shale water permeability declined with rising effective stress and such effect dropped significantly as the effective stress continues to increase.

(4) Shale permeability increased with pressure when the pressure is relatively low (less than 4 MPa in the present study), which is inconsistent with the classic Darcy's theory. This is caused by the Bingham flow (or Bingham effect) that often occurs in tiny pores.

(5) The proposed permeability model would fully capture the experimental data with reasonable accuracy in a wide range of stresses, not only for gas transport but also for water flow.

Permission to carry out fieldwork. Permission to collect shale samples was released from the local government in Kunming City, Yunnan Province.

Data accessibility. The datasets supporting this article have been uploaded as part of the electronic supplementary material [43].

Authors' contributions. M.W. collected the samples, carried out the laboratory work, participated in data analysis, carried out sequence alignments, participated in the design of the study and drafted the manuscript; D.Z. coordinated the study and critically revised the manuscript. All authors gave final approval for publication and agreed to be held accountable for the work performed therein.

Competing interests. We declare that we have no known competing interests.

Funding. This work was financially supported by the National Key R&D Programme of China, project no. 2018YFC1900200.

Acknowledgements. We gratefully thank Xiaolei Wang for his help on permeability testing of shale presented in this study.

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
