## [Peer Review File · Royal Society Open Science]

Review History

RSOS-211270.R0 (Original submission)

Review form: Reviewer 1 (Ming-zhong Gao)

Is the manuscript scientifically sound in its present form?

Yes

Are the interpretations and conclusions justified by the results?

Yes

Is the language acceptable?

Yes

Do you have any ethical concerns with this paper?

Yes

Have you any concerns about statistical analyses in this paper?

Yes

Recommendation?

Accept with minor revision (please list in comments)

Comments to the Author(s)

The topic of this study is indeed interesting and meaningful. Authors have experimentally tested the water permeability of fractured shale and come up with a verified permeability model for wide applications in the field. This manuscript is therefore recommended for publication after minor revision. My comments are as follows.

1. (Line 153) Is the sample's dimension (size) the same as that of samples used for permeability testing?
2. (Line 165) Authors should add a reference for Eq. 2 that used to calculate the tensile strength in the Brazilian Tension Test.
3. (Line 166) The units of parameters in Eq.2 should be given.
4. (Line 207) What does the deviatoric stress mean? It is not defined.
5. (Line 226) Explain the slippage effect (or Klinkenberg effect). Is it still reasonable for liquid flow in shale or not?

Review form: Reviewer 2 (Qiangui Zhang)

Is the manuscript scientifically sound in its present form?

Yes

Are the interpretations and conclusions justified by the results?

Yes

Is the language acceptable?

Yes

Do you have any ethical concerns with this paper?

No

Have you any concerns about statistical analyses in this paper?

No

Recommendation?

Accept with minor revision (please list in comments)

Comments to the Author(s)

I have carefully reviewed the manuscript entitled "Triaxial testing on water permeability evolution of fractured shale". This study, mostly based upon the experimental investigation, involves a pre-stressing strategy to artificially obtain fractured shale samples in light of the uniaxial compressive strength (UCS) and tensile strength (the development of fractures was effectively evaluated by image analysis - CT scanning), followed by triaxial testing on water permeability of the rock samples. Also, authors have extended the validity of a proposed permeability model (for gas flow, previously) to water flow in shale. Basically, the experimental design, method, procedure as well as the discussion section have met the requirement of scientific papers, and some interesting findings have also been reached. Prior to publication, several minor revisions are recommended for authors to address. Kindly see the below-listed comments I contribute.

1. Line 108, Page 5. The reason why the authors used steady-state method to measure the water permeability of shale samples should be mentioned.
2. Line 110, Page 5. Authors are supposed to add a reference for Darcy's law (Eq. 1).

3. Authors are suggested to add legends in Fig. 6 for readers to rapidly distinguish two core samples.
4. Line 133, Page 6. The sample size for NMR measurement should be mentioned. Is it consistent with the size of permeability-testing samples?
5. Line 138, Page 7. Also, the sample size for CT scanning should be mentioned.
6. The unit of Load in Fig. 8 should be printed as kN, not KN.

Review form: Reviewer 3

Is the manuscript scientifically sound in its present form?

Yes

Are the interpretations and conclusions justified by the results?

Yes

Is the language acceptable?

Yes

Do you have any ethical concerns with this paper?

No

Have you any concerns about statistical analyses in this paper?

No

Recommendation?

Accept with minor revision (please list in comments)

Comments to the Author(s)

This manuscript focuses on investigation of the water permeability evolution of fractured shale, especially the pressure and stress dependence of water permeability. The present study aims to provide theoretical guidance for evaluating and predicting the permeability of fractured shale. I would recommend publication with minor to moderate revision.

1 In Lines 19-20, "no matter how the stress condition varies", there may be a problem with the presentation.

2 Each picture should be marked with its corresponding number not only at Page 25.

3 In "4.4. CT scanning results", more discussions are needed on fracture characteristics.

4 The general language needs further editing and polishing for better readability.

Review form: Reviewer 4

Is the manuscript scientifically sound in its present form?

No

Are the interpretations and conclusions justified by the results?

Yes

Is the language acceptable?

Yes

Do you have any ethical concerns with this paper?

No

Have you any concerns about statistical analyses in this paper?

No

Recommendation?

Major revision is needed (please make suggestions in comments)

Comments to the Author(s)

This study focused on Qiongzhusi Shale to understand the evolution of shale permeability with pressure differential under varying stress conditions. The key findings of this study are, that the nanopores constitute the significant porous network, with the permeability expectedly decreasing with increasing effective stress. The shale permeability increased under low pressure conditions. The findings can have important implications towards water transport and hydraulic fracturing of these shales. Detailed imaging was used to monitor fracture development. Further, permeability model was used to match the experimental data.

The study is interesting but here are some points for consideration:

1. What is unique about your approach that is distinct from previous studies?
2. Given your approach that combines experiment, modelling and imaging, what is it that was achievable for the first time in this study?
3. What are the limitations of this study related to its application in a field setting?
4. Can you describe the shale in details with more information on mineralogy (ternary plots can be used for representation)?
5. How heterogeneous are your samples being analysed? Can the heterogeneity be correlated to the results you obtain? As an example, the differential stress may depend on the mineralogy. It might be interesting to see if the mineral brittleness index may be correlated to some of your results as well as compressibility of the shales.
6. Is it possible to characterise the pore network as to the size range and host minerals based on imaging?
7. Define the developed fractures.

Some references are missing, further comments in the attachment.

Decision letter (RSOS-211270.R0)

Dear Professor Zhang

The Editors assigned to your paper RSOS-211270 "Triaxial testing on water permeability evolution of fractured shale" have now received comments from reviewers and would like you to revise the paper in accordance with the reviewer comments and any comments from the Editors. Please note this decision does not guarantee eventual acceptance.

We invite you to respond to the comments supplied below and revise your manuscript. Below the referees' and Editors' comments (where applicable) we provide additional requirements.

Final acceptance of your manuscript is dependent on these requirements being met. We provide guidance below to help you prepare your revision.

Please submit your revised manuscript and required files (see below) no later than 21 days from today's (ie 15-Sep-2021) date. Note: the ScholarOne system will 'lock' if submission of the revision is attempted 21 or more days after the deadline. If you do not think you will be able to meet this deadline please contact the editorial office immediately.

on behalf of Prof R. Kerry Rowe (Subject Editor)
openscience@royalsociety.org

Associate Editor Comments to Author:

You've received the comments of a number of reviewers (4) that you must address not only in your manuscript but also your response to reviewers. Please be aware that we do not generally permit multiple rounds of revision, and if the reviewers are not satisfied the paper is ready for acceptance after revision, it may be rejected.

Reviewer comments to Author:

Reviewer: 1

Comments to the Author(s)

The topic of this study is indeed interesting and meaningful. Authors have experimentally tested the water permeability of fractured shale and come up with a verified permeability model for wide applications in the field. This manuscript is therefore recommended for publication after minor revision. My comments are as follows.

1. (Line 153) Is the sample's dimension (size) the same as that of samples used for permeability testing?
2. (Line 165) Authors should add a reference for Eq. 2 that used to calculate the tensile strength in the Brazilian Tension Test.
3. (Line 166) The units of parameters in Eq.2 should be given.
4. (Line 207) What does the deviatoric stress mean? It is not defined.
5. (Line 226) Explain the slippage effect (or Klinkenberg effect). Is it still reasonable for liquid flow in shale or not?

Reviewer: 2

Comments to the Author(s)

I have carefully reviewed the manuscript entitled "Triaxial testing on water permeability evolution of fractured shale". This study, mostly based upon the experimental investigation, involves a pre-stressing strategy to artificially obtain fractured shale samples in light of the uniaxial compressive strength (UCS) and tensile strength (the development of fractures was effectively evaluated by image analysis - CT scanning), followed by triaxial testing on water permeability of the rock samples. Also, authors have extended the validity of a proposed permeability model (for gas flow, previously) to water flow in shale. Basically, the experimental design, method, procedure as well as the discussion section have met the requirement of scientific papers, and some interesting findings have also been reached. Prior to publication, several minor revisions are recommended for authors to address. Kindly see the below-listed comments I contribute.

1. Line 108, Page 5. The reason why the authors used steady-state method to measure the water permeability of shale samples should be mentioned.
2. Line 110, Page 5. Authors are supposed to add a reference for Darcy's law (Eq. 1).
3. Authors are suggested to add legends in Fig. 6 for readers to rapidly distinguish two core samples.
4. Line 133, Page 6. The sample size for NMR measurement should be mentioned. Is it consistent with the size of permeability-testing samples?
5. Line 138, Page 7. Also, the sample size for CT scanning should be mentioned.
6. The unit of Load in Fig. 8 should be printed as kN, not KN.

Reviewer: 3

Comments to the Author(s)

This manuscript focuses on investigation of the water permeability evolution of fractured shale, especially the pressure and stress dependence of water permeability. The present study aims to provide theoretical guidance for evaluating and predicting the permeability of fractured shale. I would recommend publication with minor to moderate revision.

- 1 In Lines 19-20, "no matter how the stress condition varies", there may be a problem with the presentation.
- 2 Each picture should be marked with its corresponding number not only at Page 25.
- 3 In "4.4. CT scanning results", more discussions are needed on fracture characteristics.
- 4 The general language needs further editing and polishing for better readability.

Reviewer: 4

Comments to the Author(s)

This study focused on Qiongzhusi Shale to understand the evolution of shale permeability with pressure differential under varying stress conditions. The key findings of this study are, that the nanopores constitute the significant porous network, with the permeability expectedly decreasing with increasing effective stress. The shale permeability increased under low pressure conditions. The findings can have important implications towards water transport and hydraulic fracturing of these shales. Detailed imaging was used to monitor fracture development. Further, permeability model was used to match the experimental data.

The study is interesting but here are some points for consideration:

1. What is unique about your approach that is distinct from previous studies?
2. Given your approach that combines experiment, modelling and imaging, what is it that was achievable for the first time in this study?
3. What are the limitations of this study related to its application in a field setting?

4. Can you describe the shale in details with more information on mineralogy (ternary plots can be used for representation)?
5. How heterogeneous are your samples being analysed? Can the heterogeneity be correlated to the results you obtain? As an example, the differential stress may depend on the mineralogy. It might be interesting to see if the mineral brittleness index may be correlated to some of your results as well as compressibility of the shales.
6. Is it possible to characterise the pore network as to the size range and host minerals based on imaging?
7. Define the developed fractures.

Some references are missing, further comments in the attachment.

===PREPARING YOUR MANUSCRIPT===

===PREPARING YOUR REVISION IN SCHOLARONE===

Author's Response to Decision Letter for (RSOS-211270.R0)

See Appendix A.

RSOS-211270.R1

Review form: Reviewer 2 (Qiangui Zhang)

Is the manuscript scientifically sound in its present form?

Yes

Are the interpretations and conclusions justified by the results?

Yes

Is the language acceptable?

Yes

Do you have any ethical concerns with this paper?

No

Have you any concerns about statistical analyses in this paper?

No

Recommendation?

Accept as is

Comments to the Author(s)

All the comments are well modified, and in my opinion, it is suitable for publishing.

Review form: Reviewer 3

Is the manuscript scientifically sound in its present form?

Yes

Are the interpretations and conclusions justified by the results?

Yes

Is the language acceptable?

Yes

Do you have any ethical concerns with this paper?

No

Have you any concerns about statistical analyses in this paper?

No

Recommendation?

Accept as is

Comments to the Author(s)

All the modifications were checked and are accepted.

Review form: Reviewer 4

Is the manuscript scientifically sound in its present form?

Yes

Are the interpretations and conclusions justified by the results?

Yes

Is the language acceptable?

Yes

Do you have any ethical concerns with this paper?

No

Have you any concerns about statistical analyses in this paper?

No

Recommendation?

Accept as is

Comments to the Author(s)

Thank you for considering all comments very carefully and making appropriate modifications to your manuscript.

Decision letter (RSOS-211270.R1)

Dear Professor Zhang,

It is a pleasure to accept your manuscript entitled "Triaxial testing on water permeability evolution of fractured shale" in its current form for publication in Royal Society Open Science. The comments of the reviewer(s) who reviewed your manuscript are included at the foot of this letter.

on behalf of Prof R. Kerry Rowe (Subject Editor)
openscience@royalsociety.org

Reviewer comments to Author:

Reviewer: 2

Comments to the Author(s)

All the comments are well modified, and in my opinion, it is suitable for publishing.

Reviewer: 3

Comments to the Author(s)

All the modifications were checked and are accepted.

Reviewer: 4

Comments to the Author(s)

Thank you for considering all comments very carefully and making appropriate modifications to your manuscript.

Appendix A

**CHONGQING
UNIVERSITY**

SCHOOL OF RESOURCES AND SAFETY ENGINEERING
STATE KEY LABORATORY OF COAL MINE DISASTER DYNAMICS AND CONTROL

Chongqing University
No. 174 Shazheng Street, Shapingba District
Chongqing 400030, China
Tel: (+86) 023-65102421
<http://www.res.cqu.edu.cn/index.htm>

October 7, 2021

< *Royal Society Open Science* >

Editor-in-Chief: Prof. Jeremy Sanders FRS

Subject Editor (Engineering): Prof. Kerry Rowe FRS

< Triaxial testing on water permeability evolution of fractured shale > (Manuscript ID: RSOS-211270)

Response to Reviewers

Dear Editor and Reviewers:

Please find enclosed a revised version of our manuscript “Triaxial testing on water permeability evolution of fractured shale” (Manuscript Number: RSOS-211270), which we would like to resubmit for publication in the *Royal Society Open Science*.

Your comments were highly insightful and enabled us to greatly improve the quality of our manuscript. We have studied all the comments carefully and revised the manuscript accordingly. In the following pages are our point-by-point responses to each of the comments of the reviewers.

Revisions in the text are shown using **red highlights**. We hope that the revisions in the manuscript and our accompanying responses will be sufficient to make our manuscript suitable for publication in the *Royal Society Open Science*.

We deeply appreciate your critical comments concerning our manuscript. The manuscript has been resubmitted to the journal and we look forward to your positive response at your earliest convenience.

Very truly yours,
Dongming Zhang

Corresponding author: Dongming Zhang, Ph.D.
Professor, State Key Laboratory of Coal Mine Disaster Dynamics and Control
School of Resources and Safety Engineering
Chongqing University, Chongqing 400030, China
Tel.: +86 23 65111228
E-mail: zhangdm@cqu.edu.cn

Responses to the comments of Reviewer #1

The topic of this study is indeed interesting and meaningful. Authors have experimentally tested the water permeability of fractured shale and come up with a verified permeability model for wide applications in the field. This manuscript is therefore recommended for publication after minor revision. My comments are as follows.

General Response: We very much appreciate your positive comments on our work. Your comments have been addressed carefully and we hope our response could meet your approval. Kindly see below, please!

Minor Comments:

Issue 1: *(Line 153) Is the sample's dimension (size) the same as that of samples used for permeability testing?*

Response: Yes, the size of the samples used for UCS testing is same as that of samples used for permeability testing.

Issue 2: *(Line 165) Authors should add a reference for Eq. 2 that used to calculate the tensile strength in the Brazilian Tension Test.*

Response: Revision at line 173.

According to your comment, we have added a reference for Eq. 2 in the revised manuscript (also see below).

Liao ZY, Zhu JB, Tang CA. Numerical investigation of rock tensile strength determined by direct tension, Brazilian and three-point bending tests. *Int J Rock Mech Min Sci.* 2019;115: 21–32. doi:10.1016/j.ijrmms.2019.01.007

Issue 3: *The units of parameters in Eq.2 should be given.*

Response: Revision at lines 175-176.

We have added the units of parameters in Eq. 2, see below.

where σ_t is the tensile strength (N), L is the length (thickness) (m), D is the diameter (m), and P is the maximum load (N) when splitting occurs.

Issue 4: (Line 207) *What does the deviatoric stress mean? It is not defined.*

Response: Revision at line 219.

The deviatoric stress = axial stress minus confining pressure. Deviatoric stress is the stress that offsets hydrostatic stress and causes deformation.

Issue 5: (Line 226) *Explain the slippage effect (or Klinkenberg effect). Is it still reasonable for liquid flow in shale or not?*

Response: As we explained in the text, Klinkenberg effect is a non-Darcy phenomenon, where the tested apparent permeability is much higher than the intrinsic permeability. The underlying explanation for this interesting finding is that gas molecules collide extensively with pore walls instead of other gas molecules, due to the nanometer-sized pore structure of shale (extremely tight), making gas molecules “slip” on the pore walls with non-zero velocity (different with liquid flow) and, therefore, the higher permeability. Although the gas non-Darcy flow in shale has been reported to some extent in the literature, the amount of previous studies studying the Klinkenberg effect for liquid flow in shale is still low. However, there are few papers reporting that the Klinkenberg effect is still reasonable for liquid flow in shale, as listed below.

Javadpour F, McClure M, Naraghi ME. Slip-corrected liquid permeability and its effect on hydraulic fracturing and fluid loss in shale. *Fuel*. 2015;160: 549–559. doi:10.1016/j.fuel.2015.08.017

Afsharpoor A, Javadpour F. Liquid slip flow in a network of shale noncircular nanopores. *Fuel*. 2016;180: 580–590. doi:10.1016/j.fuel.2016.04.078

Responses to the comments of Reviewer #2

I have carefully reviewed the manuscript entitled “Triaxial testing on water permeability evolution of fractured shale”. This study, mostly based upon the experimental investigation, involves a pre-stressing strategy to artificially obtain fractured shale samples in light of the uniaxial compressive strength (UCS) and tensile strength (the development of fractures was effectively evaluated by image analysis - CT scanning), followed by triaxial testing on water permeability of the rock samples. Also, authors have extended the validity of a proposed permeability model (for gas flow, previously) to water flow in shale. Basically, the experimental design, method, procedure as well as the discussion section have met the requirement of scientific papers, and some interesting findings have also been reached. Prior to publication, several minor revisions are recommended for authors to address. Kindly see the below-listed comments I contribute.

General Response: We very much appreciate your positive comments.

Minor Comments:

***Issue 1:** Line 108, Page 5. The reason why the authors used steady-state method to measure the water permeability of shale samples should be mentioned.*

Response: Revision at lines 112-114.

Given that the advantages of time-saving and a straightforward solution, we used the steady-state method to measure the water permeability of shale samples, and therefore the permeability could be calculated according to Darcy’s law.

***Issue 2:** Line 110, Page 5. Authors are supposed to add a reference for Darcy’s law (Eq. 1).*

Response: Revision at line 114.

We have added a reference for Eq. 1, also see below.

Ghanizadeh A, Amann-Hildenbrand A, Gasparik M, Gensterblum Y, Krooss BM, Littke R. Experimental study of fluid transport processes in the matrix system of the European organic-rich shales: II. Posidonia Shale (Lower Toarcian, northern Germany). Int J Coal Geol. 2014;123: 20–33. doi:10.1016/j.coal.2013.06.009

Issue 3: Authors are suggested to add legends in Fig. 6 for readers to rapidly distinguish two core samples.

Response: Thanks for your helpful suggestions. We have added the legends in Fig. 6.

Figure 6. Axial stress versus strain of uniaxial compressive test.

Issue 4: Line 133, Page 6. The sample size for NMR measurement should be mentioned. Is it consistent with the size of permeability-testing samples?

Response: Revision at line 143.

Sample #1 (100 mm in height and 50 mm in diameter) was selected for NMR measurement. Its size is same as the permeability-testing samples.

Issue 5: Line 138, Page 7. Also, the sample size for CT scanning should be mentioned.

Response: Revision at line 148.

Also, sample #1 (100 mm in height and 50 mm in diameter) was selected for CT scanning.

Issue 6: The unit of Load in Fig. 8 should be printed as kN, not KN.

Response: Thanks for your helpful suggestions. We have changed “KN” to “kN” in Fig. 8.

Figure 8. The splitting curves of the Brazilian Tension Test.

Responses to the comments of Reviewer #3

This manuscript focuses on investigation of the water permeability evolution of fractured shale, especially the pressure and stress dependence of water permeability. The present study aims to provide theoretical guidance for evaluating and predicting the permeability of fractured shale. I would recommend publication with minor to moderate revision.

General Response: First of all, thanks for your positive comments. Additionally, your minor comments have been addressed carefully. Kindly see below, please.

Minor comments:

Issue 1: *In Lines 19-20, “no matter how the stress condition varies”, there may be a problem with the presentation.*

Response: We have removed the sentence: “no matter how the stress condition varies”.

Issue 2: *Each picture should be marked with its corresponding number not only at Page 25.*

Response: According to the requirement of the journal, figure captions must be listed at the end of the paper and cannot be in the text. We ask for your understanding regarding this issue. Thank you!

Issue 3: *In “4.4. CT scanning results”, more discussions are needed on fracture characteristics.*

Response: Revision at lines 203-209.

We have added more discussions on fracture characteristics: “While figure 11 shows the slices images and three-dimensional reconstruction model of shale after fracturing treatment. It can be clearly seen from the figure that many obvious fractures appear at the same slice position compared with figure 10. After three-dimensional reconstruction, we find that three penetrating fractures have been formed in the sample. These three main developed fractures in shale as the

dominant pathways for fluid (water) transport shows that our pre-stressing strategy to create more reliable fractures in shale for permeability measurement has met the requirement”.

Issue 4: The general language needs further editing and polishing for better readability.

Response: Thank you for this comment. We have carefully reviewed and edited the manuscript to improve the coherence, cohesion, precision and grammatical range of the manuscript.

Responses to the comments of Reviewer #4

This study focused on Qiongzhusi Shale to understand the evolution of shale permeability with pressure differential under varying stress conditions. The key findings of this study are, that the nanopores constitute the significant porous network, with the permeability expectedly decreasing with increasing effective stress. The shale permeability increased under low pressure conditions. The findings can have important implications towards water transport and hydraulic fracturing of these shales. Detailed imaging was used to monitor fracture development. Further, permeability model was used to match the experimental data. The study is interesting but here are some points for consideration:

General Response: We sincerely appreciate your time and work on our study and thanks a lot for giving us the opportunity to revise our manuscript following your suggestions. From the scientific & academic point of view, your comments are indeed both high-quality and constructive that help us improve our study. Kindly see our following one-by-one responses to each of your comments and we truly hope our response could meet your approval. Thanks a million!

***Issue 1:** What is unique about your approach that is distinct from previous studies?*

Response: We developed a pre-stressing strategy to artificially create fractures inside the shale samples, according to the uniaxial compressive strength and tensile strength. While previous studies often used a single pseudo-fracture by splitting the shale sample to measure the permeability. Compared with previous studies, the random fractures produced by our approach are more in line with the engineering practice. Also, our approach would alleviate the challenge that shale permeability is often hard to measure due to its fine-grained property.

***Issue 2:** Given your approach that combines experiment, modelling and imaging, what is it that was achievable for the first time in this study?*

Response: What we believe to be the most contributing outcome for the first time in this study based upon our approach, is the artificially-created fractures in shale for permeability testing. The fracture patterns would, to the most extent, replicate the in-situ scenario, since the underlying mechanisms for fracturing (propagation and development) are same, i.e., under the effect of stress. Also, we extended the application scope of a permeability model to further capture the water permeability evolution characteristics in fractured shale.

Issue 3: What are the limitations of this study related to its application in a field setting?

Response: From the scientific point of view, the limitation is that there are still several issues to overcome, to measure the real-time permeability after hydraulic fracturing. Because current technologies (also for this study) cannot do this due to the lack of associated theory and technology. For example, how to directly test the post-fracturing water (or gas) permeability individually.

Issue 4: Can you describe the shale in details with more information on mineralogy (ternary plots can be used for representation)?

Response: Revision at lines 90-93.

Thank you for raising this constructive comment, as mineralogy is indeed important when shale is in contact with water. We have tested the mineralogical composition using X-ray diffraction (XRD) analysis and added a few sentences in the revised manuscript. Also, we ask for your understanding that we cannot plot a ternary cartoon due to the insufficient samples. Instead, we have added a table (Table 1) to describe the mineralogical composition of the shale sample.

Table 1 lists the tested mineralogical composition of the shale sample using X-ray diffraction (XRD) analysis. Both the previous data and table 1 demonstrate the huge potential for promising gas reserves in southwest China, especially the Qiongzhusi Formation.

Table 1. Mineralogical composition (%) of the Qiongzhusi shale sample.

Sample	Quartz	Dolomite	Feldspar	Pyrite	Calcite	Clay	
						Illite	Chlorite
	38.3	7.5	5.5	2.6	7.8	29.9	8.4

Issue 5: *How heterogeneous are your samples being analysed? Can the heterogeneity be correlated to the results you obtain? As an example, the differential stress may depend on the mineralogy. It might be interesting to see if the mineral brittleness index may be correlated to some of your results as well as compressibility of the shales.*

Response: We very much appreciate your professional comment. We fully agree with you that the mineral brittleness index would be correlated to the properties related to permeability, such rock compressibility, rock brittleness and so on. However, those exactly fall with the area of rock mechanics, while our interests are fluid flow in porous media. Thus, what you mentioned goes beyond the scope of our area, and we are not able to, at this time, to find the correlation between the differential stress and mineralogy. Anyway, we thank you for this constructive comment and will try to involve it in our future work. As for the heterogeneity, we drilled the samples from an entire rock mass and, therefore, the heterogeneity could be ignored in this study.

Issue 6: *Is it possible to characterise the pore network as to the size range and host minerals based on imaging?*

Response: Currently, the resolution of CT imaging mostly remains at the micro-meter scale, which is valid for capturing the characteristic of fractures, not pore network. In this study, the pre-stressing strategy for creating fractures in shale for permeability measurement was, therefore, evaluated by CT scanning. However, as for the pore network and host minerals, it should be noted that CT alone would not effectively capture the smallest features of shale, e.g., nanopores

and organic matter, in terms of precisely capturing porosity, due to the relatively low CT imaging resolution (Arif et al., 2021).

Arif, M., Mahmoud, M., Zhang, Y., Iglauer, S., 2021. X-ray tomography imaging of shale microstructures: A review in the context of multiscale correlative imaging. *Int. J. Coal Geol.* 233, 103641. <https://doi.org/10.1016/j.coal.2020.103641>

Issue 7: Define the developed fractures.

Response: Revision at line 132.

The developed fractures mean that the fracture extends, and a relatively-developed pore network and randomly-distributed cracks have been formed.

Issue 8: Some references are missing, further comments in the attachment.

Response: We very much appreciate your extensive work and time on our work. Your comments in the attachment are indeed high-quality and truly help us to improve our study. We have carefully studied all your comments and revised them accordingly. The only one we are not able to address is the comment for section 2: geological setting. This is due to the lack of insufficient published literature introducing the geological background of our study. Anyway, we tried our best to specifically describe it as much as possible in the text. Moreover, as for the selection of samples, we thank you for your mentioned reference (Basu et al., *J. Mar. Sci. Eng.* 2020, 8(2), 136; <https://doi.org/10.3390/jmse8020136>). As this study is already finished, we will pay attention to this suggestion in our future experiments.

Additionally, we listed all the newly-added references below for you to check. Thanks a million!

Gao S, Dong D, Tao K, Guo W, Li X, Zhang S. Experiences and lessons learned from China's shale gas development: 2005 – 2019. *J Nat Gas Sci Eng.* 2021;85: 103648.

Ma L, Slater T, Doweij PJ, Yue S, Rutter EH, Taylor KG, et al. Hierarchical integration of porosity in shales. *Sci Rep.* 2018;8: 11683. doi:10.1038/s41598-018-30153-x

Caineng Z, Dazhong D, Shejiao W, Jianzhong L, Xinjing L, Yuman W, et al. Geological characteristics and resource potential of shale gas in China. *Pet Explor Dev.* 2010;37: 641–653. doi:10.1016/S1876-3804(11)60001-3

Jia B, Jin L, Mibeck BAF, Smith SA, Sorensen JA. An integrated approach of measuring permeability of naturally fractured shale. *J Pet Sci Eng.* 2020;186: 106716. doi:10.1016/j.petrol.2019.106716

Zhang X, Shi W, Hu Q, Zhai G, Wang R, Xu X. Pressure – dependent fracture permeability of marine shales in the Northeast Yunnan area, Southern China. *Int J Coal Geol.* 2019;214: 103237. doi:10.1016/j.coal.2019.103237

An C, Killough J, Xia X. Investigating the effects of stress creep and effective stress coefficient on stress-dependent permeability measurements of shale rock. *J Pet Sci Eng.* 2021;198: 108155. doi:10.1016/j.petrol.2020.108155

Cui G, Liu J, Wei M, Shi R, Elsworth D. Why shale permeability changes under variable effective stresses: New insights. *Fuel.* 2018;213: 55–71. doi:10.1016/j.fuel.2017.10.068

Liao ZY, Zhu JB, Tang CA. Numerical investigation of rock tensile strength determined by direct tension, Brazilian and three-point bending tests. *Int J Rock Mech Min Sci.* 2019;115: 21–32. doi:10.1016/j.ijrmms.2019.01.007